# Random Forests Machine Learning Applied to PEER Structural Performance Experimental Columns Database

**Konstantinos G. Megalooikonomou** [1] and **Grigorios N. Beligiannis** [1,2,*]

1 School of Science and Technology, Hellenic Open University, Parodos Aristotelous 18, 26335 Patras, Greece; std153412@ac.eap.gr

2 Department of Food Science & Technology, University of Patras, Agrinio Campus, G. Seferi 2, 30100 Agrinio, Greece

* Correspondence: gbeligia@upatras.gr

**Abstract:** Columns play a very important role in structural performance and, therefore, this paper contributes to the critical need for failure mode prediction of reinforced concrete (RC) columns by exploring the capabilities of random forest machine learning (ML) based on a well-known experimental column database. Known as the PEER structural performance database, it assembles the results of over 400 cyclic, lateral-load tests of reinforced concrete columns. The database describes tests of spiral or circular hoop-confined columns, rectangular tied columns and columns with or without lap splices of longitudinal reinforcement at the critical sections. The efficiency towards the aforementioned goal of supervised ML methods such as random forests using a randomly assigned test set from the Pacific Earthquake Engineering Research Center (PEER) database is examined here. The overall accuracy score for rectangular RC columns is 94% and for circular RC columns is 86%. The latter performances are influenced by the size of the testing and training sets of data and are independent of the number of decision trees in the forest employed in the random forest algorithm. The performances of random forests in postdicting the failure mode of RC columns prove that ML has great promise in revolutionizing the profession of earthquake engineering.

**Keywords:** reinforced concrete; columns; PEER structural performance database; machine learning; random forests; failure mode

## 1. Introduction

Machine learning (ML) is a subfield of artificial intelligence (AI) and an advanced form of data analysis and computation that employs the high elaboration speed and pattern recognition techniques of computers for knowledge output from data. In other words, it is a computer programming technique inspired by AI that allows computers to improve their learning abilities through data supplies or data access. This resembles the way human beings improve their intelligence in real life. There are four generalized categories of ML. To be more specific, there is supervised learning, semi-supervised learning, unsupervised learning and reinforcement learning. In supervised learning, the desired output is known by the trainer, where the trainer is the human being that can ascribe physical meaning to the data and characterize it by adding a tag or correcting system errors. The machine is trained based on inputs with tags that are connected to a corresponding output. Through this process, the machine develops a predictive model for the connection of this input to a certain output. This does not differ from the way that knowledge is learned in a classroom, with a teacher available to correct any errors.

The mode of failure of structural members, such as reinforced concrete columns, depends on several factors, such as their geometric characteristics, the longitudinal reinforcement, the efficiency of confinement through the transverse reinforcement and the loading history. Their behavior throughout the loading range is controlled by competing mechanisms of resistance such as flexure, shear, buckling of longitudinal bars when they

are subjected to compressive loads and, in the case of lap splices, the lap splice mechanism of the development of reinforcing bars. Very often, a combination of such mechanisms characterizes the macroscopic behavior of the column, especially in cases of cyclic load reversals. Various predictive models have been developed in the past to determine both the strength as well as the deformation capacity of the columns, with the uncertainty being at least one order of magnitude greater in terms of deformation capacity rather than strength, as evidenced by comparisons with test results [1,2].

System identification and damage detection is a twofold area that utilizes ML to imitate a structural system and predict its deterministic seismic response. Laboratory tests of reinforced concrete (RC) structures have provided one source of data that enables ML methods to identify their failure modes, strength, capacities and constitutive behaviors [3]. ML methods in which the algorithms are used to learn from the data have been used recently for risk assessment and prediction models in civil engineering [4–11]. In this regard, some studies have focused on predicting failure modes and shear strength for beam–column joints [6,9,12,13]. For instance, Mitra et al. (2011) [12] categorized non-ductile joint shear failure versus ductile beam yielding failure for interior beam–column joints. And, Tang et al., 2022 [14] examined the design and application of a low-cycle reciprocating loading test on 23 recycled aggregate concrete-filled steel tube columns and 3 ordinary concrete-filled steel tube columns. Moreover, in the latter study, they applied artificial intelligence to estimate the influence of parameter variation on the seismic performance of concrete columns. Specifically, random forests with hyperparameters tuned by the firefly algorithm were chosen. Similar studies with multi-objective optimization analyses are included in [15].

In this paper, a supervised learning algorithm called the random forest is tested as a predictive model for the first time for its performance in postdicting the failure mode of RC columns against a widely used experimental database originally assembled by Berry and Eberhard (2004) [16]. Known as the PEER structural performance database, it assembles the results of over 400 cyclic, lateral-load tests of reinforced concrete columns. The database describes tests of spiral or circular hoop-confined columns, rectangular tied columns and columns with or without lap splices of longitudinal reinforcement at the critical sections. For each test, where the information is available, the database provides the column geometry, column material properties, column reinforcing details, test configuration (including P-Delta configuration), axial load, digital lateral force displacement history at the top of the column and top displacement that preceded various damage observations.

This paper has the following contributions in the research area of ML methods in earthquake engineering:

- According to the authors' knowledge, the PEER structural performance database is employed for the first time in order to detect the failure mode of RC columns.
- Rectangular RC columns are examined for the first time for their failure mode detection through the random forest ML method [3,17].
- The influence of the main design variables on the column ductility and failure mode is also thoroughly examined.
- Finally, all the performance metrics necessary for the evaluation of the ML methodology in detecting the failure mode of RC columns are provided too.

The structure of this study is the following: after the introduction which describes the initiatives of this research paper, the employed data and the performed methodology are described in Section 2. In the latter section, the influence of the main design variables on the column ductility and failure mode is given in detail, along with the statistical representation of the database. The steps of the performed supervised ML method in Python programming language are provided here, too. Finally, the output results along with their discussion are presented in Section 4, while the conclusions and future work are presented in Section 5.

## 2. Materials and Methods

Statistics of the aforementioned PEER structural performance database are provided below for the column depth, aspect ratio, axial load ratio, longitudinal reinforcement ratio ($\rho_l$) and transverse reinforcement ratio ($\rho_s$) [1,2].

### 2.1. Statistical Representation of the PEER Structural Performance Database

Table 1 provides the mean values (Mean), Standard deviation (*Std*) and Coefficient of variation (*CoV*) of key column properties for 306 rectangular reinforced columns and 177 spiral reinforced columns. Statistics are provided for the column depth, aspect ratio, axial-load ratio, longitudinal reinforcement ratio ($\rho_l$) and transverse reinforcement ratio ($\rho_s$).

**Table 1.** Column property statistics.

| Column Property | Rectangular Reinforced (306 Tests) | | | Spiral Reinforced (177 Tests) | | |
|---|---|---|---|---|---|---|
| | Mean | Std | CoV | Mean | Std | CoV |
| Depth (mm) | 323.43 | 116.5 | 0.36 | 420.97 | 202.11 | 0.48 |
| Aspect Ratio | 3.44 | 1.44 | 0.42 | 3.31 | 1.96 | 0.59 |
| Axial-Load Ratio | 0.27 | 0.19 | 0.73 | 0.14 | 0.14 | 1.04 |
| $\rho_l$ (%) | 2.45 | 1.00 | 0.41 | 2.62 | 1.02 | 0.39 |
| $\rho_s$ (%) | 1.34 | 1.07 | 0.80 | 0.93 | 0.74 | 0.80 |

### 2.2. Influence of Main Design Variables to Column Ductility and Failure Mode

One important goal in seismic structural assessment procedures is the reliable estimation of the available capacity of structural members for inelastic deformation, as well as their available ductility. Ductility drives assessment since its magnitude underlies the general design philosophy (i.e., through the q-μ-T relationships it controls the magnitude of strength reduction from the elastic demands that may be tolerated before failure) and, in current code practice (EN 1998-1 2004 [18] and AASHTO LRFD 2013 [19], FEMA 440 2005 [20]), its magnitude is reflected on the specific reinforcing requirements of members and structures.

In the experimental database report of Berry and Eberhard (2004) [12], the nominal column failure mode was classified as (a) flexure critical, (b) flexure–shear critical, or (c) shear critical, according to the following criteria:

- If no shear damage was reported by the experimentalist, the column was classified as flexure critical.
- If shear damage (diagonal cracks) was reported, the absolute maximum effective force ($F_{eff}$: absolute maximum measured force in the experimental column response) was compared with the calculated "ideal" force corresponding to a maximum axial compressive strain in the concrete cover set equal to 0.004, which corresponds to the spalling of unconfined concrete ($F_{0.004}$). The failure displacement ductility at an effective force equal to 80% maximum $\mu_{fail}$ was determined from the experimental envelope. If the maximum effective force $F_{eff} < 0.95 \cdot F_{0.004} F_{eff} < 0.95 \cdot F_{0.004} F_{eff} < 0.95 \cdot F_{0.004}$ or if the failure displacement ductility was less than or equal to 2 ($\mu_{fail} \leq 2 \mu_{fail} \leq 2 \mu_{fail} \leq 2$), the column was classified as shear critical. Otherwise, the column was classified as flexure–shear critical. All columns in the database are divided into two sub-groups according to cross-sectional shape (rectangular and circular section columns).

In this section, the displacement ductility value clouds—as defined by the reported experimental responses—are correlated against important design parameters and plotted in graphs to illustrate the parametric dependencies of this variable on the column failure mode.

For example, considering the concrete strength, the following points are made: (a) higher strength materials are marked by lower ultimate strain, (b) strain can be enhanced through confinement, (c) a higher concrete strength results in a lower compression zone both at yielding and at failure. In general, it can be said that higher concrete strength causes a reduction in ductility. This finding is confirmed by both groups of rectangular tied columns and by the spiral reinforced columns, as can be seen in Figures 1 and 2. For the spiral reinforced columns, it is more clearly evident that the ductility is increased for specimens with lower concrete strengths.

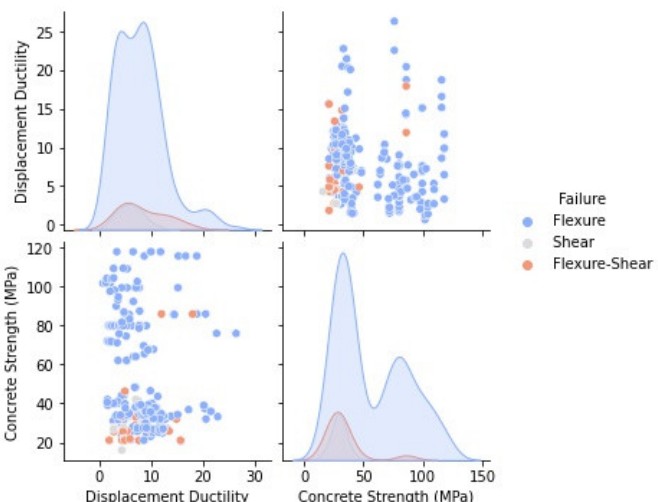

**Figure 1.** Effect of concrete strength on displacement ductility for the rectangular reinforced columns.

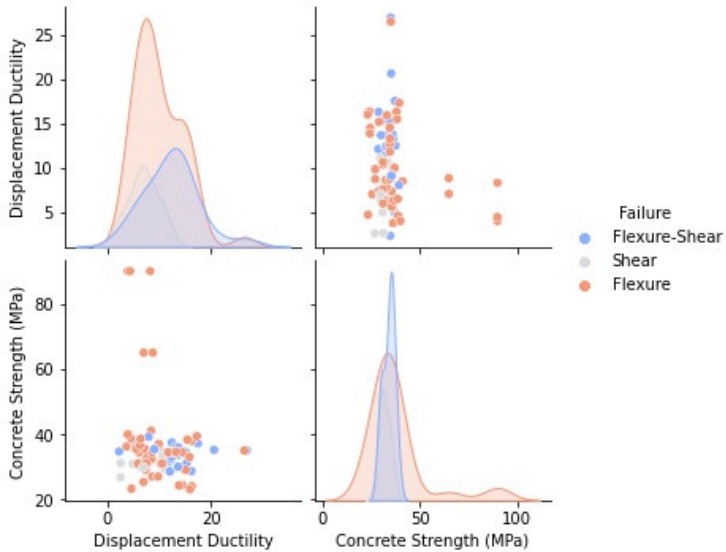

**Figure 2.** Effect of concrete strength on displacement ductility for the spiral reinforced columns.

During the flexural analysis of a section both at yielding and at failure, the presence of a compressive axial load increases the depth of the compressive zone, as compared to an identical section without axial force. Based on the above remark, the presence of the compressive axial load reduces the curvature ductility of a section. The experimental data confirm this tendency, with brittleness being more evident in the cases where the axial load ratios exceeded the point of balanced failure (see Figures 3 and 4).

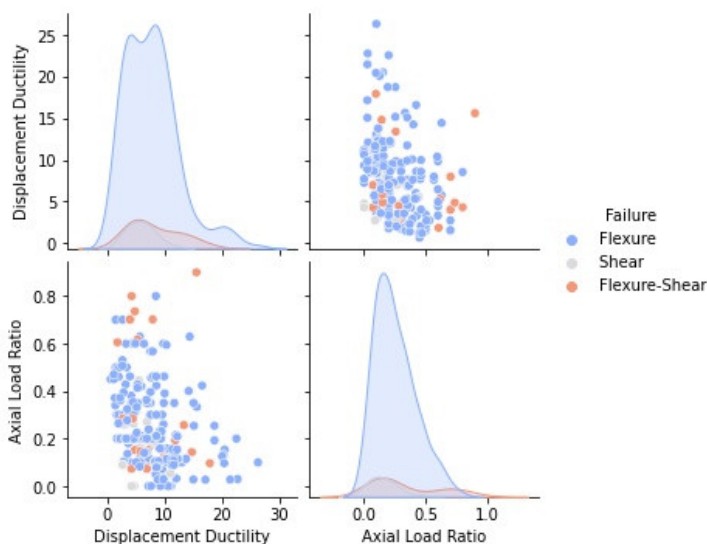

**Figure 3.** Effect of axial load ratio on displacement ductility for the rectangular reinforced columns.

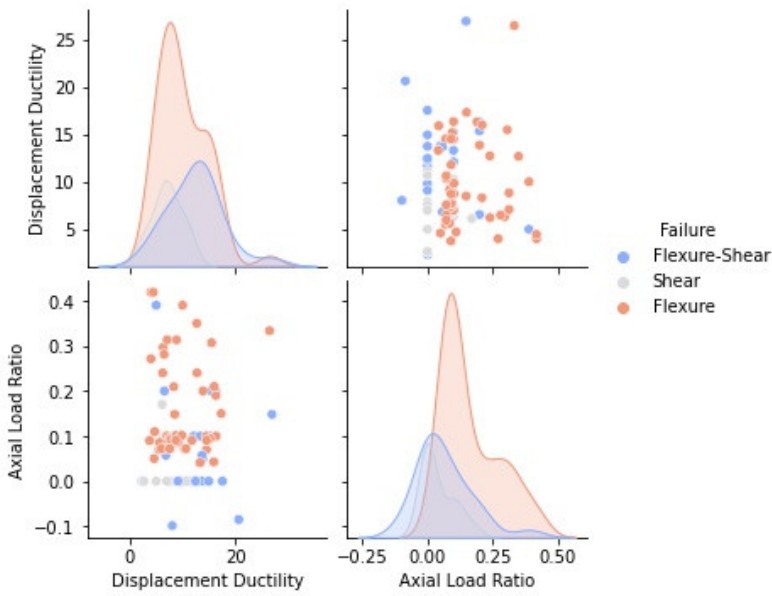

**Figure 4.** Effect of axial load ratio on displacement ductility for the spiral reinforced columns.

The shear span to depth ratio, known as the aspect ratio, $a = L_s/h$, has a determining influence on the characteristics of shear behavior. In a column of small shear span to depth ratio, shear deformation may become appreciable, compared with flexural deformation. A dominant shear response causes a more pronounced pinching in the force-deformation (hysteresis) curve and a faster degradation of the hysteresis energy dissipation capacity. Interestingly, the experimental data show that the ductility ratio increases with a decreasing aspect ratio (Figures 5 and 6); this perplexing result is attributed to the fact that the yield displacement increases at a quadratic rate with shear span length $L_s$, whereas the ultimate displacement is linear with $L_s$ and thus the ductility estimate is inversely proportional to $L_s/h$ or $a$. The following expressions relate the flexural component of column response with aspect ratio, illustrating the source of the observations interpreting the experimental trend:

$$\text{-Yield Curvature}: \ \varphi_y = 2.1 \cdot \frac{\varepsilon_{sy}}{h} \tag{1}$$

$$\text{-Yield Displacement}: \ \Delta_y = \frac{1}{3} \cdot \varphi_y \cdot L_s^2 \approx \frac{2}{3} \cdot \varepsilon_{sy} \cdot \frac{L_s}{h} \cdot L_s = \frac{2}{3} \cdot \varepsilon_{sy} \cdot a \cdot L_s \tag{2}$$

$$\begin{aligned}\text{-Ultimate Displacement}: \ \ \Delta_u &\approx \Delta_y + \varphi_{pl} \cdot \ell_{pl} \cdot L_s = \Delta_y + \frac{\varepsilon_{pl}}{\frac{2h}{3}} \cdot \ell_{pl} \cdot L_s \\ &= \Delta_y + 1.5\varepsilon_{pl} + \ell_{pl} \cdot a\end{aligned} \tag{3}$$

$$\text{-Displacement Ductility}: \ \mu_\Delta \approx 1 + 2.3(\mu_\varepsilon - 1) \cdot \frac{\ell_{pl}}{L_s} \tag{4}$$

where $\ell_{pl}$ is the plastic hinge length (approximated as $0.5h$ in practical calculations), $\varepsilon_{pl}$ the nonlinear (past yielding) part of the tension reinforcement total strain and $\mu_\varepsilon$ is the required bar strain ductility.

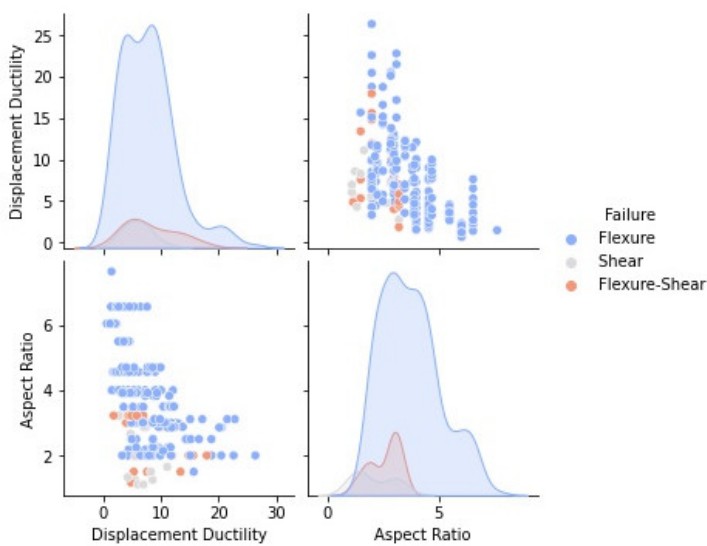

**Figure 5.** Effect of aspect ratio on displacement ductility for the rectangular reinforced columns.

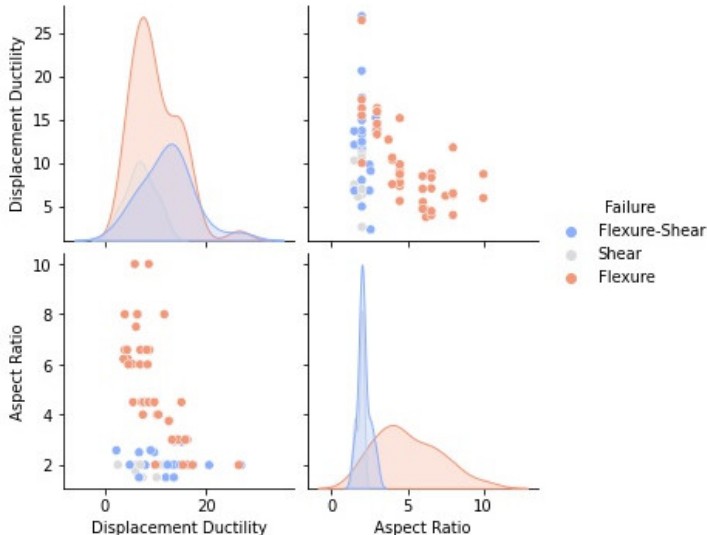

**Figure 6.** Effect of aspect ratio on displacement ductility for the spiral reinforced columns.

Figures 7 and 8 depict the relationship between the maximum shear stress (maximum experimental shear force divided by the gross area of the column) normalized by the square root of the concrete strength of each column and the associated displacement ductility. Columns with a higher ductility also support a higher shear force as both parameters are correlated to the same variable, i.e., the quality and quantity of detailing. The observation is also consistent with the trends of Figures 5 and 6 which illustrate that displacement ductility is inversely proportional to aspect ratio, which, in turn, for a given member flexural resistance, is inversely proportional to shear demand (since $V_{Ed} = M_{Ed}/(h \cdot a)$).

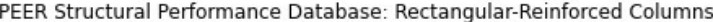

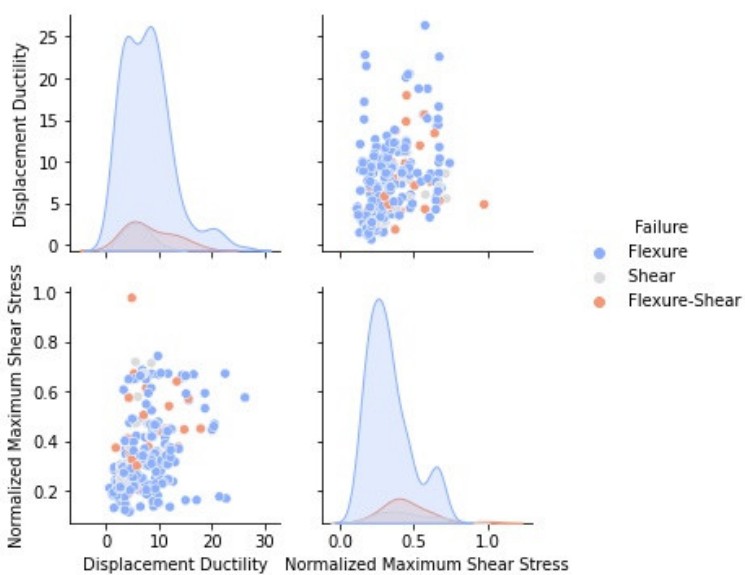

**Figure 7.** Maximum shear stress vs. displacement ductility for the rectangular reinforced columns.

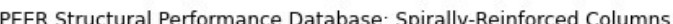

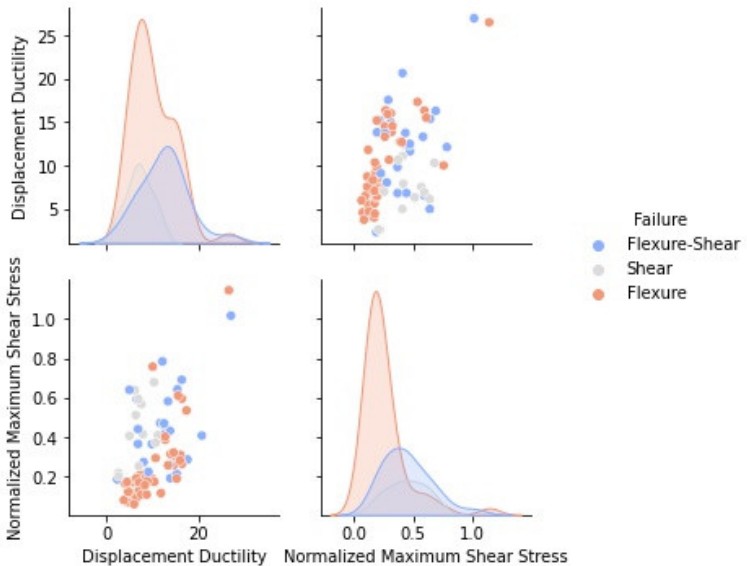

**Figure 8.** Maximum shear stress vs. displacement ductility for the spiral reinforced columns.

The database trends are also examined with reference to lateral confinement—which is generally acknowledged to enhance the deformation capacity of the column. The arrangement of confining reinforcement is important in this regard; a column with closely spaced

stirrups and well-distributed longitudinal reinforcement shows very little strength decay even when being subjected to very high axial forces with magnitudes exceeding the limit of balanced failure. The plotted trends confirm this general expectation: the displacement ductility increases with the transverse reinforcement ratio, as shown in Figures 9 and 10.

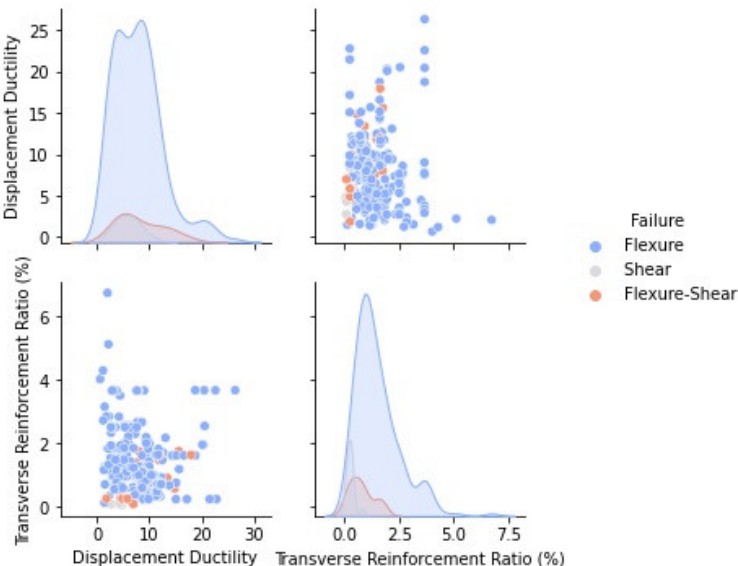

**Figure 9.** Effect of transverse reinforcement ratio on displacement ductility for the rectangular reinforced columns.

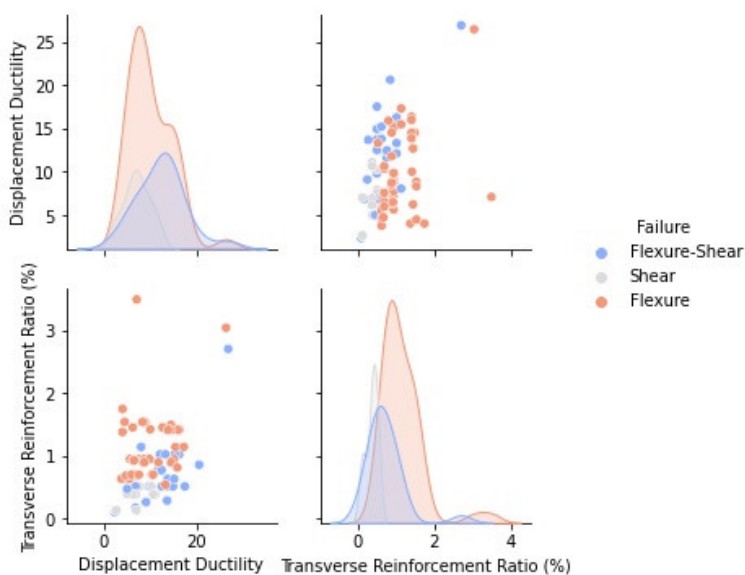

**Figure 10.** Effect of transverse reinforcement ratio on displacement ductility for the spiral reinforced columns.

### 2.3. Supervised ML-Based Prediction of Column Failure Mode with Random Forests

At this point after the statistical description of the available experimental data and the examination of the influence of the main design variables to the displacement ductility of the columns along with their failure mode, it is time to introduce the methodology of the failure mode prediction of reinforced concrete columns by exploring the capabilities

of ML methods. The procedure working towards the aforementioned goal of supervised ML methods, such as random forests, using a randomly assigned test set from the PEER database is described thoroughly here.

Random forests are an ensemble method and are based on the construction of many different decision trees [21]. Every decision tree alone cannot provide an effective prediction but all together can be a more effective model. This is, therefore, the essence of ensemble methods. That is, to create models that result from the combination of many algorithms of which each one apart is not sufficiently effective.

Random forests are proposed in order to confront the overfitting problem, where decision trees are insufficient. Overfitting is a result of a very well-fitted model to the training data (the collected observations). The fitting is so effective that the models' predictions to new data are not satisfactory. The random forest algorithm creates a set of different decision trees, each one with different characteristics that obtains the average value of the output or the resulting value of the majority of the decision trees and therefore can be considered as a majority voting algorithm. The creation of different decision trees with different characteristic sets to each tree is called bagging and it is a subcategory of ensemble methods. Another random source of the random forest is the selection of the characteristics in each tree node. There are many hyperparameters that need to be defined for the application of random forest algorithms, such as:

- The estimator number that defines the number of decision trees.
- The maximum feature number that defines the maximum feature number during the separation of nodes in each decision tree.
- The maximum depth: the maximum depth in each decision tree.
- Minimum sample points at each node separation: the minimum sample point number that should be taken into account at each node.

Likewise with decision trees, random forests do not demand any preprocessing. Moreover, they are less sensitive to overfitting in comparison to decision trees. However, random forests are slower in learning compared to decision trees with many hyperparameters. Finally, due to the fact that random forests are random, there is not a full certainty for their results since the latter could be changed.

Random Forests with Python [22]

In any machine learning problem, the following steps are taken:

1. The question is set and the demanded data are defined.
2. The data are obtained in an accessible form.
3. Any lack of data or uncertainty is defined and corrected accordingly.
4. The data are prepared for the machine learning model.
5. A baseline model is set that is intended to be overcome.
6. The model is trained with the training data.
7. Model predictions are made with the test data.
8. The predictions are compared to the known test goals and the performance metrics are computed.
9. If the performance is not satisfactory, we adjust the model and obtain more data or another modeling technique is tested too.

In the following section, the results of the application of the above-described methodology (see also the flowchart in Figure 11) are given based on Python programming language and the performance metrics are provided, too. It is shown that the classification of the columns based on the latter ML method is accurate in identifying the failure mode of the collected experimental data. It should be underlined that in the following results, the random state in the random forests is set to value 42, which means that the results will be the same every time that splitting of the data to training and testing data is performed for reproducible results (random_state = 42 means that no matter how many times the

code is executed, the result would be the same, i.e., the same values in training and testing datasets). Finally, the number of the trees in random forests is set to 1000.

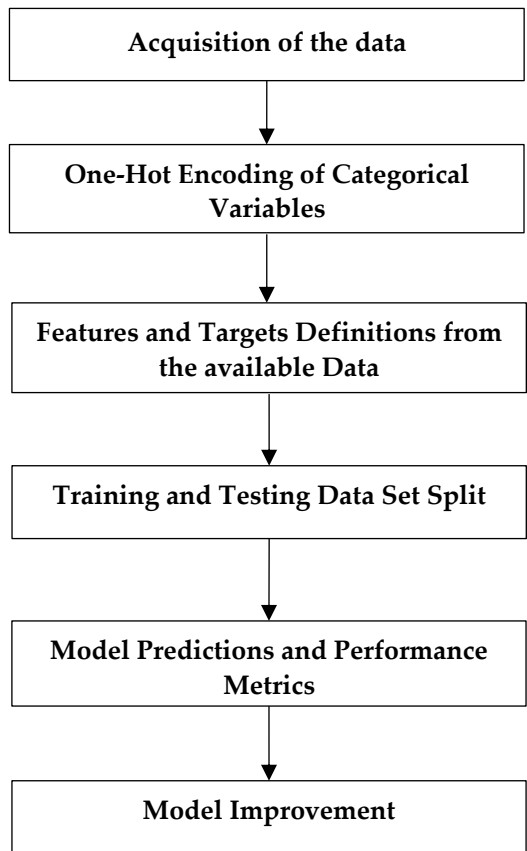

**Figure 11.** Flowchart of the proposed methodology with random forest supervised ML method.

Before presenting the results of the above-described methodology, a description of the sensitivity of the previously described hyperparameters is necessary. More details with results of this sensitivity will be given in the next section. Regarding the number of estimators, it should be underlined that more trees should be able to produce a more generalized result but by choosing a greater number of trees, the time complexity of the random forest model also increases. In addition, the maximum depth of a tree in the random forest is defined as the longest path between the root node and the leaf node. As the maximum depth of the decision tree increases, the performance of the model over the training dataset increases continuously. The same is valid for the test dataset but with a certain limit over which it decreases rapidly. In the proposed methodology, the maximum depth of the tree is selected so that nodes are expanded until all leaves are pure or until all leaves contain less than the minimum samples split. The default value of the latter hyperparameter (minimum samples split) is two and so this is the minimum number of samples required to split an internal node that was defined in the same way in the proposed methodology. However, by increasing the value of this hyperparameter, the number of splits that happen in the decision tree can be reduced and therefore prevent the model from overfitting. Finally, the maximum number of features is the maximum features provided to each tree in a random forest or else the number of features to consider when looking for the best split. It is a good convention to consider the default value of this parameter, which is one.

## 3. Results

After examination of the entire PEER database for circular and rectangular RC columns, it can be seen that the necessary parameters to define the control variables that affect the

mode of failure and displacement ductility, as described in Section 2.2, are not available for all the columns of the database. Therefore, firstly the data files defining all the necessary parameters like aspect ratio, axial load ratio, concrete strength, transverse reinforcement ratio and normalized maximum shear stress (see Section 2.2) are generated and are divided into two groups according to the shape of the section, i.e., rectangular and circular RC columns (see Appendix A).

The above-mentioned data are divided into training data and into test data for each column section type. The proportion of the dataset to include in the testing data split is defined as 25% and the training data size is automatically set to the complement of the testing data size. During training, the model is allowed to see the correct answers, in this case the failure mode of the RC columns (flexure, flexure–shear and shear), so that it can learn how to predict the failure mode from the provided features. As described previously, it is anticipated that there is a connection between all the features and the failure mode goal and the model should try to figure out this connection. After this step, the model is asked during testing to predict the failure mode of the test data having access only to the features data and not the correct answers of the failure mode. Since these answers are available to the supervisors, the accuracy of the model can be examined. Generally, when a model is trained, the random data are divided into training data and test data in order for the trainer to have a representation of the whole available data.

### 3.1. Rectangular RC Columns

Below, the performance metrics for the case of the rectangular RC columns are provided. It can be seen that random forests have 94% accuracy in predicting the actual failure mode of the columns of the tested data. This accuracy score could be explained based on Table 2 by dividing the sum of the diagonal matrix terms with the sum of all the terms of the table. More performance metrics are provided in Table 3. It should be noted that by examining separately each of the influencing parameters included in the features data, the most crucial for the model's success is the transverse reinforcement ratio which confirms that the model correctly figured out the connection between all the features and the failure mode goal. This is crucial for establishing a physical meaning-based ML method prediction model. Finally, it should be also underlined based on Table 3 that the model is more successful in predicting the flexural and shear modes of failure compared to the other one. This makes sense since flexure-shear is more difficult also in the real engineering world to be detected in terms of seismic assessment. Finally, it should be clear that Table 2 clarifies the conception of Figure 12 and Table 3 does the same for Table 2.

**Table 2.** Confusion matrix in numbers for ML prediction of the failure mode of rectangular RC columns of PEER structural performance database with random forest method.

|  |  | **Confusion Matrix in Numbers *** | | |
|---|---|---|---|---|
|  | Flexure | 55 | 2 | 0 |
| True Values | Flexure–Shear | 2 | 2 | 0 |
|  | Shear | 0 | 0 | 1 |
|  |  | Flexure | Flexure–Shear | Shear |
|  |  | | Predicted Values | |

* See also Figure 12.

**Table 3.** Performance metrics.

| | **Performance Metrics *** | | | | | | |
|---|---|---|---|---|---|---|---|
| | **True Positive** | **True Negative** | **False Positive** | **False Negative** | **Accuracy** | **Precision** | **Recall** |
| Flexure | 55 | $2 + 1 + 0 + 0 = 3$ | $2 + 0 = 2$ | $2 + 0 = 2$ | $(55 + 3)/(55 + 3 + 2 + 2) = 58/62 = 94\%$ | $(55)/(55 + 2) = 55/57 = 97\%$ | $(55)/(55 + 2) = 55/57 = 97\%$ |

**Table 3.** *Cont.*

| | True Positive | True Negative | False Positive | False Negative | Accuracy | Precision | Recall |
|---|---|---|---|---|---|---|---|
| | | | | | **Performance Metrics ***| | |
| Flexure–Shear | 2 | 55 + 0 + 0 + 1= 56 | 2 + 0 = 2 | 2 + 0 = 2 | (2 + 56)/(2 + 56 + 2 + 2) = 58/62 = 94% | (2)/(2 + 2) = 2/4 = 50% | (2)/(2 + 2) = 2/4 = 50% |
| Shear | 1 | 55 + 2 + 2 + 2 = 61 | 0 + 0 = 0 | 0 + 0 = 0 | (1 + 61)/(1 + 61 + 0 + 0) = 62/62 = 100% | (1)/(1 + 0) = 1/1 = 100% | (1)/(1 + 0) = 1/1 = 100% |

\* See also Table 2.

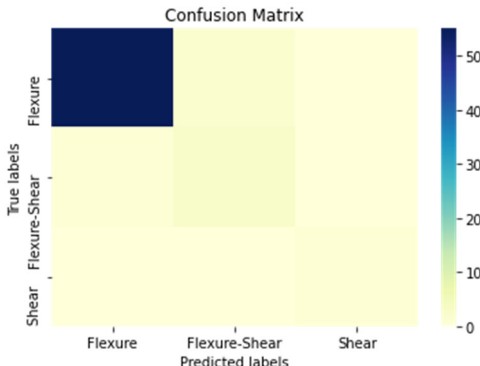

**Figure 12.** Confusion matrix as performance metric for ML prediction of the failure mode of rectangular RC columns of PEER structural performance database with random forest method.

*3.2. Circular RC Columns*

Below, the performance metrics for the case of the circular RC columns are provided.

It can be seen that random forests have 86% accuracy in predicting the actual failure mode of the columns of the tested data. This accuracy score could be explained based on Table 4 by dividing the sum of diagonal terms with the sum of all the terms of the table. More performance metrics are provided in Table 5. It should be noted that by examining separately each of the influencing parameters included in the features data, the most crucial for the model's success is the transverse reinforcement ratio which confirms that the model correctly figured out the connection between all the features and the failure mode goal. This is crucial for establishing a physical meaning-based ML method prediction model. Finally, it should be also underlined based on Table 5 that the model in the case of circular RC columns is more successful in predicting the flexural and flexural–shear modes of failure, compared to the other one, which makes sense since brittle failures further demand nonlinear structural analyses to be deterministically detected. Finally, it should be clear that Table 4 clarifies the conception of Figure 13 and Table 5 does the same for Table 4.

**Table 4.** Confusion matrix in numbers for ML prediction of the failure mode of circular RC columns of PEER structural performance database with random forest method.

| | | **Confusion Matrix in Numbers *** | | |
|---|---|---|---|---|
| True Values | Flexure | 12 | 0 | 0 |
| | Flexure–Shear | 1 | 5 | 1 |
| | Shear | 0 | 1 | 1 |
| | | Flexure | Flexure–Shear | Shear |
| | | Predicted Values | | |

\* See also Figure 13.

**Table 5.** Performance metrics.

| | True Positive | True Negative | False Positive | False Negative | Accuracy | Precision | Recall |
|---|---|---|---|---|---|---|---|
| | | | **Performance Metrics \*** | | | | |
| Flexure | 12 | 5 + 1 + 1 + 1 = 8 | 1 + 0 = 1 | 0 + 0 = 0 | (12 + 8)/(12 + 8 + 1 + 0) = 20/21 = 95% | (12)/(12 + 1) = 12/13 = 92% | (12)/(12 + 0) = 12/12 = 100% |
| Flexure–Shear | 5 | 12 + 0 + 0 + 1 = 13 | 0 + 1 = 1 | 1 + 1 = 2 | (5 + 13)/(5 + 13 + 1 + 2) = 18/21 = 95% | (5)/(5 + 1) = 5/6 = 83% | (5)/(5 + 2) = 5/7 = 71% |
| Shear | 1 | 12 + 0 + 1 + 5 = 18 | 0 + 1 = 1 | 0 + 1 = 1 | (1 + 18)/(1 + 18 + 1 + 1) = 19/21 = 90% | (1)/(1 + 1) = 1/2 = 50% | (1)/(1 + 1) = 1/2 = 50% |

\* See also Table 4.

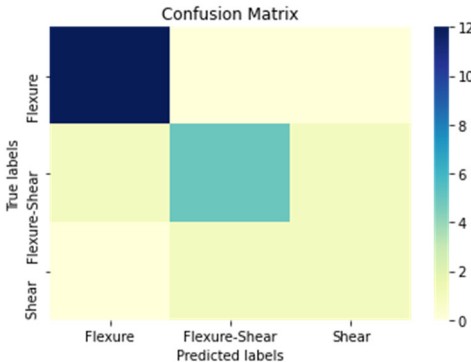

**Figure 13.** Confusion matrix as performance metric for ML prediction of the failure mode of circular RC columns of PEER structural performance database with random forest method.

### 3.3. Parametric Sensitivity of Random Forest Algorithm

As described already, the proportion of the dataset to include in the testing data split is defined as 25% and the training data size is automatically set to the complement of the testing data size. Moreover, the estimator number that defines the number of decision trees in the forest is set to 1000. Here, the parametric sensitivity of these two parameters on the accuracy performance score of random forests in postdicting the failure mode of RC columns will be examined. The following Figures depict this sensitivity and it can be seen that the number of decision trees in the forest has no influence on the confusion matrix of the performance of the random forest algorithm (Figures 14 and 15). Finally, as seems reasonable by increasing the testing set data (and decreasing the training data at the same time), there is a decrease in the accuracy score of rectangular RC columns and less-so in circular RC columns (Figures 16 and 17).

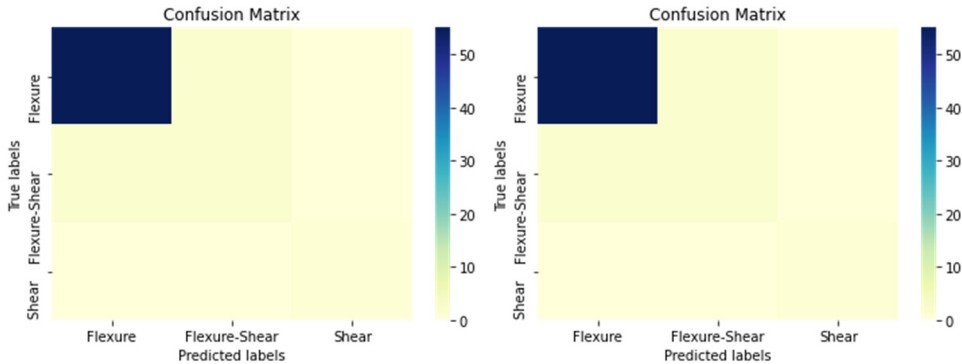

**Figure 14.** Parametric sensitivity of the number of decision trees in the forest on the accuracy score of ML prediction of the failure mode of rectangular RC columns of PEER structural performance database with random forest method (100 estimators left—94% accuracy score, 10,000 estimators right—94% accuracy score).

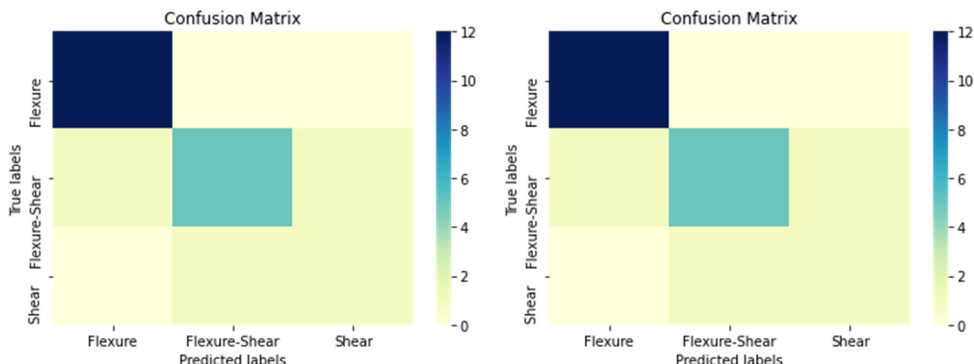

**Figure 15.** Parametric sensitivity of the number of decision trees in the forest on the accuracy score of ML prediction of the failure mode of circular RC columns of PEER structural performance database with random forest method (100 estimators left—86% accuracy score, 10,000 estimators right—86% accuracy score).

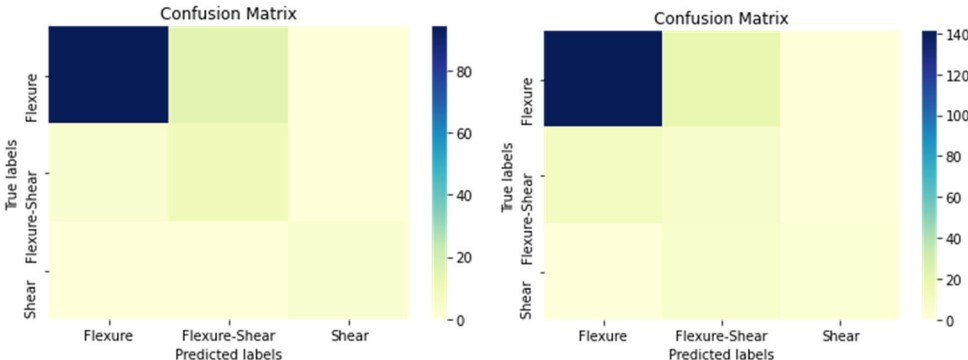

**Figure 16.** Parametric sensitivity of the size of the testing set data on the accuracy score of ML prediction of the failure mode of rectangular RC columns of PEER structural performance database with random forest method (50% testing set data left—85% accuracy score, 75% testing set data right—81% accuracy score).

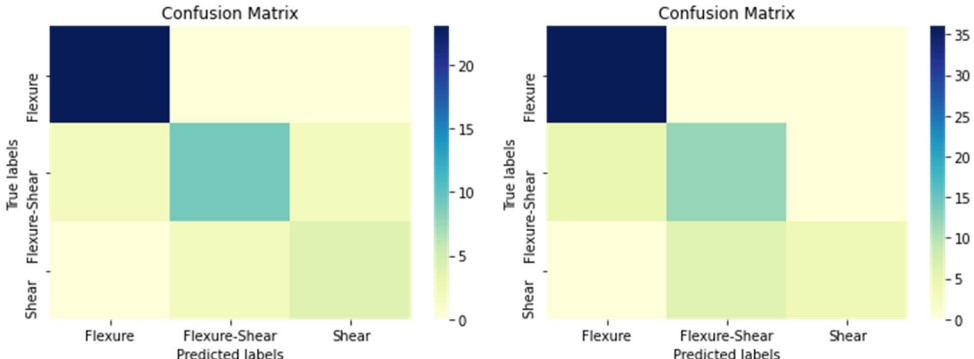

**Figure 17.** Parametric sensitivity of the size of the testing set data on the accuracy score of ML prediction of the failure mode of circular RC columns of PEER structural performance database with random forest method (50% testing set data left—86% accuracy score, 75% testing set data right—83% accuracy score).

## 4. Discussion

The state of the art in modeling the lateral load response of columns leaves a lot to be desired: improved response estimation of the behavior of columns that are susceptible to shear failure after flexural yielding; better procedures to estimate shear strength and the pattern of degradation thereof with increasing displacement ductility; the need to

account for reinforcement pullout and its effects on stiffness; the shape of the hysteresis loops; the detrimental effects of axial load at large displacement limits; and the magnitude of deformation (drift ratio) associated with milestone events in the response curve of the column member. These are open issues that need to be settled before the performance-based assessment framework may be considered complete and dependable [1,2,23–30]. In this framework, it is evident from the current study that incorporating physical knowledge (through experimental databases) in ML methods can accurately predict the failure mode of RC columns. From the above-described results, random forests are successful in predicting the failure mode of RC columns, both circular and rectangular, especially the more ductile ones (flexural failure) and, moreover, in attributing the failure mode to the most crucial column features, like the transverse reinforcement ratio. The overall accuracy score for rectangular RC columns is 94% and for circular RC columns is 86%. The latter performances are influenced by the size of the testing and training sets of data and are independent of the number of decision trees in the forest employed in the random forest algorithm. Finally, the low precision and recall scores for brittle failures, especially for circular RC columns, are confirmed also by other studies [3], where it is suggested that brittle failures are crucial in governing the retrofitting and operational strategies of critical infrastructures to adopt other supervised ML methods, such as Neural Networks and Deep Learning.

## 5. Conclusions

The prediction of the failure mode of RC columns is crucial in defining the retrofit solutions of buildings and bridges in the modern world. Current strategies include nonlinear structural analysis procedures, which demand a lot of effort and time in order to be performed accurately. This study explores the capability of incorporating physical knowledge into ML methods for predicting the failure mode of RC columns. To this end, the PEER structural performance database is employed and the influence of main design variables on the column ductility and failure mode are examined. It can be seen that supervised ML methods, such as random forests, using a randomly assigned test set from the PEER database and incorporating physical knowledge into them can classify columns' failure modes accurately, proving that ML has great promise in revolutionizing the profession of earthquake engineering. Finally, according to the authors' knowledge and the state of the art, the PEER structural performance database is employed for the first time in research in order to identify columns' failure modes through supervised ML methods. The section of the column is also a variable that was not considered thoroughly and recent results refer only to circular columns. This study will be the basis for further examination of other supervised ML methods in detecting RC columns' failure modes, such as Decision Trees, k-Nearest Neighbor, Neural Networks and Deep Learning.

**Author Contributions:** K.G.M. conducted the data analysis, comparison and interpretation. G.N.B. supervised, reviewed and validated the proposed methodology. All authors have read and agreed to the published version of the manuscript.

**Funding:** This research was partially conducted with financial support from the Alexander S. Onassis Public Benefit Foundation which provided KGM with a triennial scholarship (Scholarship Code: F ZI 086-1 2012-2013/01/09/2012–29/02/2016) to pursue a Ph.D. degree in Civil Engineering at the University of Cyprus.

**Informed Consent Statement:** Not applicable.

**Data Availability Statement:** Data are contained within the article.

**Acknowledgments:** The first author would like to thank the Alexander S. Onassis Public Benefit Foundation, whose financial support is greatly appreciated. The present research work is partially based on the PhD thesis of KGM [1], which was extended to the application of supervised ML methods to the postdiction of reinforced concrete columns' failure modes.

**Conflicts of Interest:** The authors declare no conflict of interest.

**Appendix A**

Random forests grow many classification trees. To classify a new object from an input vector, the input vector is given to each of the trees in the forest. Each tree gives a classification, and the tree "votes" for that class. The forest chooses the classification with the most votes (over all the trees in the forest). In the following tables, the feature values and the true label values—showing the classification nature of the algorithm—are given.

**Table A1.** Feature values and true label values of circular RC columns from PEER structural performance database.

| Feature Values | | | | | True Label Values (1 = Flexure, 2 = Flexure–Shear, 3 = Shear) |
|---|---|---|---|---|---|
| *Aspect Ratio* | *Axial Load Ratio* | $\rho_s$ *(%)* | $f_c$ *(MPa)* | $v_{max}/\sqrt{fc}$ | *Failure* |
| 2.00 | 0.00 | 0.51 | 37.5 | 0.42 | 2 |
| 2.00 | 0.00 | 0.51 | 37.2 | 0.29 | 2 |
| 2.50 | 0.00 | 0.51 | 36 | 0.37 | 2 |
| 2.00 | 0.00 | 0.51 | 30.6 | 0.42 | 3 |
| 2.00 | 0.00 | 0.76 | 31.1 | 0.47 | 2 |
| 1.50 | 0.00 | 0.51 | 30.1 | 0.57 | 3 |
| 2.00 | 0.00 | 0.38 | 29.5 | 0.41 | 3 |
| 2.00 | 0.20 | 1.02 | 28.7 | 0.69 | 2 |
| 2.00 | 0.20 | 1.02 | 31.2 | 0.64 | 2 |
| 2.00 | 0.20 | 0.51 | 29.9 | 0.59 | 2 |
| 1.50 | 0.10 | 1.02 | 28.6 | 0.78 | 2 |
| 2.00 | 0.10 | 1.02 | 36.2 | 0.58 | 2 |
| 2.00 | 0.00 | 0.51 | 33.7 | 0.43 | 2 |
| 2.00 | 0.00 | 0.51 | 34.8 | 0.31 | 2 |
| 2.00 | 0.10 | 0.51 | 33.4 | 0.51 | 3 |
| 2.50 | 0.10 | 0.51 | 34.3 | 0.44 | 2 |
| 1.50 | 0.10 | 0.51 | 35 | 0.68 | 3 |
| 1.50 | 0.10 | 0.38 | 34.4 | 0.59 | 3 |
| 1.75 | 0.17 | 0.38 | 36.7 | 0.64 | 3 |
| 2.00 | 0.00 | 0.38 | 33.2 | 0.37 | 3 |
| 2.00 | 0.00 | 0.39 | 30.9 | 0.41 | 3 |
| 2.00 | 0.00 | 0.76 | 32.3 | 0.47 | 2 |
| 2.00 | 0.00 | 0.77 | 33.1 | 0.47 | 2 |
| 2.00 | 0.19 | 1.42 | 38 | 0.60 | 1 |
| 2.00 | 0.39 | 0.47 | 37 | 0.64 | 2 |
| 2.00 | 0.39 | 1.42 | 37 | 0.76 | 1 |
| 6.22 | 0.05 | 0.63 | 38.8 | 0.07 | 1 |
| 6.22 | 0.09 | 0.63 | 36.2 | 0.08 | 1 |
| 2.93 | 0.05 | 0.63 | 35.9 | 0.19 | 2 |
| 2.92 | 0.10 | 0.63 | 34.4 | 0.21 | 2 |
| 7.50 | 0.24 | 1.45 | 34.5 | 0.18 | 1 |
| 3.75 | 0.24 | 1.45 | 34.5 | 0.39 | 1 |
| 3.75 | 0.35 | 1.45 | 34.5 | 0.40 | 1 |
| 6.01 | 0.07 | 0.63 | 35.8 | 0.12 | 1 |
| 3.01 | 0.07 | 1.49 | 34.3 | 0.31 | 1 |
| 3.00 | 0.10 | 1.41 | 24.1 | 0.27 | 1 |
| 3.00 | 0.21 | 1.41 | 23.1 | 0.31 | 1 |
| 6.00 | 0.10 | 0.68 | 25.4 | 0.13 | 1 |
| 3.00 | 0.10 | 1.41 | 24.4 | 0.26 | 1 |
| 3.00 | 0.20 | 1.41 | 24.3 | 0.32 | 1 |
| 6.00 | 0.11 | 0.68 | 23.3 | 0.13 | 1 |

**Table A1.** *Cont.*

| Feature Values | | | | | True Label Values (1 = Flexure, 2 = Flexure–Shear, 3 = Shear) |
|---|---|---|---|---|---|
| *Aspect Ratio* | *Axial Load Ratio* | *$\rho_s$ (%)* | *$f_c$ (MPa)* | *$v_{max}/\sqrt{fc}$* | *Failure* |
| 4.50 | 0.09 | 0.94 | 29 | 0.19 | 1 |
| 4.50 | 0.09 | 0.94 | 29 | 0.19 | 1 |
| 4.50 | 0.09 | 0.94 | 35.5 | 0.17 | 1 |
| 4.50 | 0.09 | 0.94 | 35.5 | 0.21 | 1 |
| 4.50 | 0.09 | 0.94 | 35.5 | 0.18 | 1 |
| 4.50 | 0.09 | 0.94 | 32.8 | 0.19 | 1 |
| 4.50 | 0.09 | 0.94 | 32.8 | 0.17 | 1 |
| 4.50 | 0.09 | 0.94 | 32.5 | 0.18 | 1 |
| 4.50 | 0.10 | 0.94 | 27 | 0.20 | 1 |
| 4.50 | 0.10 | 0.94 | 27 | 0.19 | 1 |
| 4.50 | 0.10 | 0.94 | 27 | 0.19 | 1 |
| 1.50 | 0.06 | 0.28 | 30 | 0.26 | 2 |
| 1.50 | 0.06 | 0.17 | 30 | 0.37 | 2 |
| 6.00 | 0.15 | 0.89 | 41.1 | 0.19 | 1 |
| 1.99 | 0.31 | 1.14 | 38.3 | 0.61 | 1 |
| 1.99 | -0.10 | 1.14 | 39.2 | 0.28 | 2 |
| 1.99 | 0.15 | 1.14 | 39.4 | 0.54 | 1 |
| 1.99 | 0.15 | 2.70 | 35 | 1.02 | 2 |
| 1.99 | -0.08 | 0.85 | 35.2 | 0.41 | 2 |
| 1.99 | 0.33 | 3.04 | 35 | 1.14 | 1 |
| 8.00 | 0.30 | 0.92 | 36.6 | 0.19 | 1 |
| 8.00 | 0.27 | 1.38 | 40 | 0.17 | 1 |
| 8.00 | 0.28 | 0.92 | 38.6 | 0.19 | 1 |
| 4.00 | 0.07 | 0.70 | 31 | 0.18 | 1 |
| 8.00 | 0.07 | 0.70 | 31 | 0.09 | 1 |
| 10.00 | 0.07 | 0.70 | 31 | 0.06 | 1 |
| 4.00 | 0.07 | 0.70 | 31 | 0.11 | 1 |
| 4.00 | 0.07 | 0.70 | 31 | 0.30 | 1 |
| 3.00 | 0.09 | 0.89 | 34.5 | 0.32 | 1 |
| 8.00 | 0.09 | 0.89 | 34.5 | 0.12 | 1 |
| 10.00 | 0.09 | 0.89 | 34.5 | 0.11 | 1 |
| 3.00 | 0.04 | 0.54 | 34.6 | 0.26 | 1 |
| 3.00 | 0.04 | 0.81 | 33 | 0.28 | 1 |
| 6.58 | 0.31 | 1.54 | 65 | 0.18 | 1 |
| 6.58 | 0.31 | 3.49 | 65 | 0.17 | 1 |
| 6.58 | 0.42 | 1.74 | 90 | 0.17 | 1 |
| 6.58 | 0.21 | 1.54 | 90 | 0.16 | 1 |
| 6.58 | 0.42 | 1.54 | 90 | 0.17 | 1 |
| 2.58 | 0.00 | 0.10 | 34.7 | 0.19 | 2 |
| 2.58 | 0.00 | 0.26 | 35.4 | 0.23 | 2 |
| 2.00 | 0.00 | 0.13 | 29.8 | 0.25 | 3 |
| 2.00 | 0.00 | 0.13 | 26.8 | 0.22 | 3 |
| 2.00 | 0.00 | 0.13 | 31.2 | 0.20 | 3 |

**Table A2.** Feature values and true label values of rectangular RC columns from PEER structural performance database.

| Feature Values | | | | | True Label Values (1 = Flexure, 2 = Flexure–Shear, 3 = Shear) |
|---|---|---|---|---|---|
| *Aspect Ratio* | *Axial Load Ratio* | $\rho_s$ *(%)* | $f_c$ *(MPa)* | $v_{max}/\sqrt{fc}$ | *Failure* |
| 2.18 | 0.26 | 1.50 | 23.1 | 0.48 | 1 |
| 2.18 | 0.21 | 2.30 | 41.4 | 0.42 | 1 |
| 2.18 | 0.42 | 2.00 | 21.4 | 0.48 | 1 |
| 2.18 | 0.60 | 3.50 | 23.5 | 0.47 | 1 |
| 4.00 | 0.38 | 2.83 | 23.6 | 0.25 | 1 |
| 4.00 | 0.21 | 2.22 | 25 | 0.21 | 1 |
| 4.00 | 0.10 | 0.86 | 46.5 | 0.18 | 1 |
| 4.00 | 0.30 | 1.22 | 44 | 0.26 | 1 |
| 4.00 | 0.30 | 0.80 | 44 | 0.26 | 1 |
| 4.00 | 0.30 | 0.57 | 40 | 0.26 | 1 |
| 4.00 | 0.22 | 1.56 | 28.3 | 0.25 | 1 |
| 4.00 | 0.39 | 1.99 | 40.1 | 0.27 | 1 |
| 4.00 | 0.50 | 0.66 | 41 | 0.29 | 1 |
| 4.00 | 0.50 | 0.32 | 40 | 0.29 | 1 |
| 4.00 | 0.70 | 1.26 | 42 | 0.29 | 1 |
| 4.00 | 0.70 | 0.70 | 39 | 0.30 | 1 |
| 4.00 | 0.70 | 2.33 | 40 | 0.31 | 1 |
| 4.00 | 0.20 | 2.55 | 25.6 | 0.21 | 1 |
| 4.00 | 0.20 | 2.55 | 25.6 | 0.21 | 1 |
| 4.00 | 0.20 | 2.55 | 25.6 | 0.22 | 1 |
| 4.00 | 0.20 | 2.55 | 25.6 | 0.21 | 1 |
| 3.00 | 0.10 | 1.70 | 32 | 0.23 | 1 |
| 3.00 | 0.10 | 1.70 | 32 | 0.24 | 1 |
| 3.00 | 0.30 | 2.08 | 32.1 | 0.36 | 1 |
| 3.00 | 0.30 | 2.08 | 32.1 | 0.36 | 1 |
| 2.97 | 0.10 | 2.17 | 26.9 | 0.32 | 1 |
| 1.50 | 0.33 | 1.18 | 20.6 | 0.57 | 1 |
| 1.50 | 0.17 | 0.81 | 21.6 | 0.47 | 3 |
| 1.50 | 0.35 | 1.39 | 21 | 0.61 | 2 |
| 4.00 | 0.03 | 0.32 | 24.8 | 0.15 | 1 |
| 4.00 | 0.03 | 0.32 | 24.8 | 0.14 | 1 |
| 4.00 | 0.03 | 0.32 | 24.8 | 0.14 | 1 |
| 2.00 | 0.14 | 0.57 | 32 | 0.45 | 2 |
| 2.00 | 0.15 | 0.57 | 29.9 | 0.51 | 2 |
| 1.65 | 0.05 | 0.36 | 27.1 | 0.45 | 3 |
| 2.00 | 0.80 | 0.73 | 21.1 | 0.58 | 2 |
| 2.00 | 0.80 | 0.73 | 21.1 | 0.61 | 1 |
| 2.00 | 0.90 | 1.75 | 21.1 | 0.57 | 2 |
| 3.00 | 0.70 | 0.73 | 28.8 | 0.41 | 2 |
| 3.00 | 0.70 | 0.73 | 28.8 | 0.40 | 2 |
| 3.00 | 0.70 | 1.75 | 28.8 | 0.38 | 2 |
| 3.00 | 0.11 | 0.38 | 27.9 | 0.25 | 1 |
| 3.00 | 0.11 | 0.38 | 27.9 | 0.24 | 1 |
| 3.00 | 0.11 | 0.38 | 27.9 | 0.25 | 1 |
| 3.00 | 0.12 | 0.38 | 24.8 | 0.27 | 1 |
| 3.00 | 0.11 | 0.38 | 27.9 | 0.25 | 1 |
| 3.00 | 0.11 | 0.38 | 27.9 | 0.23 | 1 |
| 1.25 | 0.18 | 0.21 | 31.8 | 0.71 | 3 |

**Table A2.** *Cont.*

| Feature Values | | | | | True Label Values (1 = Flexure, 2 = Flexure–Shear, 3 = Shear) |
|---|---|---|---|---|---|
| *Aspect Ratio* | *Axial Load Ratio* | $\rho_s$ (%) | $f_c$ (MPa) | $v_{max}/\sqrt{fc}$ | *Failure* |
| 1.25 | 0.45 | 0.21 | 33 | 0.72 | 3 |
| 2.50 | 0.40 | 1.61 | 85.7 | 0.66 | 1 |
| 2.50 | 0.63 | 1.61 | 85.7 | 0.65 | 1 |
| 2.50 | 0.63 | 1.61 | 85.7 | 0.67 | 1 |
| 2.50 | 0.25 | 1.61 | 115.8 | 0.59 | 1 |
| 2.50 | 0.25 | 1.61 | 115.8 | 0.59 | 1 |
| 2.50 | 0.42 | 1.61 | 115.8 | 0.67 | 1 |
| 2.50 | 0.42 | 1.61 | 115.8 | 0.67 | 1 |
| 1.50 | 0.26 | 0.91 | 25.8 | 0.64 | 2 |
| 1.50 | 0.62 | 0.91 | 25.8 | 0.67 | 2 |
| 2.00 | 0.35 | 0.50 | 99.5 | 0.66 | 1 |
| 2.00 | 0.35 | 0.75 | 99.5 | 0.66 | 1 |
| 2.00 | 0.35 | 0.61 | 99.5 | 0.69 | 1 |
| 2.00 | 0.35 | 0.50 | 99.5 | 0.65 | 1 |
| 2.00 | 0.35 | 0.50 | 99.5 | 0.65 | 1 |
| 2.00 | 0.35 | 0.50 | 99.5 | 0.67 | 1 |
| 2.00 | 0.35 | 0.50 | 99.5 | 0.65 | 1 |
| 1.16 | 0.74 | 0.89 | 46.3 | 0.98 | 2 |
| 2.88 | 0.12 | 0.33 | 34.7 | 0.36 | 2 |
| 2.88 | 0.12 | 0.33 | 34.7 | 0.37 | 1 |
| 2.88 | 0.15 | 0.48 | 26.1 | 0.44 | 2 |
| 2.88 | 0.15 | 0.48 | 26.1 | 0.41 | 1 |
| 2.88 | 0.11 | 0.33 | 33.6 | 0.35 | 2 |
| 2.88 | 0.11 | 0.33 | 33.6 | 0.39 | 1 |
| 2.88 | 0.07 | 0.33 | 33.6 | 0.33 | 3 |
| 2.88 | 0.07 | 0.33 | 33.6 | 0.35 | 1 |
| 2.88 | 0.11 | 0.67 | 33.4 | 0.38 | 2 |
| 2.88 | 0.11 | 0.67 | 33.4 | 0.37 | 1 |
| 2.88 | 0.11 | 1.47 | 33.5 | 0.45 | 2 |
| 2.88 | 0.11 | 1.47 | 33.5 | 0.45 | 1 |
| 2.88 | 0.11 | 0.92 | 33.5 | 0.45 | 2 |
| 2.88 | 0.11 | 0.92 | 33.5 | 0.45 | 1 |
| 5.50 | 0.10 | 1.54 | 29.1 | 0.12 | 1 |
| 5.50 | 0.09 | 0.93 | 30.7 | 0.12 | 1 |
| 5.50 | 0.10 | 1.54 | 29.2 | 0.12 | 1 |
| 5.50 | 0.10 | 0.93 | 27.6 | 0.15 | 1 |
| 5.50 | 0.20 | 1.54 | 29.4 | 0.15 | 1 |
| 5.50 | 0.18 | 0.93 | 31.8 | 0.14 | 1 |
| 5.50 | 0.26 | 1.54 | 33.3 | 0.15 | 1 |
| 5.50 | 0.27 | 0.93 | 32.4 | 0.15 | 1 |
| 5.50 | 0.28 | 1.54 | 31 | 0.16 | 1 |
| 5.50 | 0.27 | 0.93 | 31.8 | 0.15 | 1 |
| 1.11 | 0.16 | 0.28 | 34.9 | 0.58 | 3 |
| 1.98 | 0.16 | 0.31 | 34.9 | 0.47 | 3 |
| 1.11 | 0.27 | 0.28 | 42 | 0.67 | 3 |
| 1.50 | 0.10 | 0.26 | 29.9 | 0.42 | 3 |
| 3.00 | 0.21 | 2.19 | 39.3 | 0.36 | 1 |
| 3.00 | 0.31 | 1.26 | 39.8 | 0.37 | 1 |
| 2.86 | 0.00 | 0.85 | 43.6 | 0.34 | 1 |

**Table A2.** *Cont.*

| Feature Values | | | | | True Label Values (1 = Flexure, 2 = Flexure–Shear, 3 = Shear) |
|---|---|---|---|---|---|
| *Aspect Ratio* | *Axial Load Ratio* | $\rho_s$ (%) | $f_c$ (MPa) | $v_{max}/\sqrt{fc}$ | *Failure* |
| 2.86 | 0.14 | 1.69 | 34.8 | 0.38 | 1 |
| 2.86 | 0.15 | 2.54 | 32 | 0.47 | 1 |
| 2.86 | 0.13 | 1.95 | 37.3 | 0.46 | 1 |
| 2.86 | 0.13 | 1.95 | 39 | 0.45 | 1 |
| 4.56 | 0.30 | 1.22 | 80 | 0.23 | 1 |
| 4.56 | 0.30 | 1.22 | 80 | 0.22 | 1 |
| 4.56 | 0.20 | 1.22 | 80 | 0.18 | 1 |
| 4.56 | 0.20 | 1.22 | 80 | 0.25 | 1 |
| 4.56 | 0.20 | 1.83 | 80 | 0.25 | 1 |
| 4.56 | 0.30 | 1.83 | 80 | 0.23 | 1 |
| 4.56 | 0.30 | 1.83 | 80 | 0.23 | 1 |
| 4.56 | 0.20 | 1.83 | 80 | 0.20 | 1 |
| 4.56 | 0.20 | 3.66 | 80 | 0.18 | 1 |
| 4.56 | 0.30 | 3.66 | 80 | 0.23 | 1 |
| 4.56 | 0.20 | 3.66 | 80 | 0.24 | 1 |
| 4.56 | 0.30 | 3.66 | 80 | 0.24 | 1 |
| 4.56 | 0.20 | 1.22 | 80 | 0.31 | 1 |
| 4.56 | 0.30 | 1.22 | 80 | 0.30 | 1 |
| 4.56 | 0.30 | 1.22 | 80 | 0.31 | 1 |
| 4.56 | 0.20 | 1.22 | 80 | 0.37 | 1 |
| 4.56 | 0.20 | 1.83 | 80 | 0.29 | 1 |
| 4.56 | 0.20 | 1.83 | 80 | 0.35 | 1 |
| 4.56 | 0.30 | 1.83 | 80 | 0.31 | 1 |
| 4.56 | 0.30 | 1.83 | 80 | 0.31 | 1 |
| 4.56 | 0.20 | 3.66 | 80 | 0.31 | 1 |
| 4.56 | 0.20 | 3.66 | 80 | 0.31 | 1 |
| 4.56 | 0.30 | 3.66 | 80 | 0.30 | 1 |
| 4.56 | 0.30 | 3.66 | 80 | 0.32 | 1 |
| 3.83 | 0.10 | 0.37 | 27.2 | 0.30 | 1 |
| 3.83 | 0.24 | 0.37 | 27.2 | 0.33 | 1 |
| 3.83 | 0.09 | 0.48 | 28.1 | 0.31 | 1 |
| 3.83 | 0.23 | 0.48 | 28.1 | 0.35 | 1 |
| 3.22 | 0.09 | 0.08 | 26.9 | 0.26 | 3 |
| 3.22 | 0.07 | 0.08 | 33.1 | 0.20 | 2 |
| 3.22 | 0.28 | 0.08 | 25.5 | 0.29 | 2 |
| 3.22 | 0.26 | 0.08 | 27.6 | 0.30 | 3 |
| 3.22 | 0.26 | 0.25 | 27.6 | 0.32 | 3 |
| 3.22 | 0.09 | 0.08 | 26.9 | 0.25 | 3 |
| 3.22 | 0.07 | 0.08 | 33.1 | 0.19 | 2 |
| 3.22 | 0.28 | 0.25 | 25.5 | 0.35 | 2 |
| 2.00 | 0.10 | 3.67 | 76 | 0.58 | 1 |
| 2.00 | 0.20 | 3.67 | 76 | 0.67 | 1 |
| 2.00 | 0.10 | 3.67 | 86 | 0.46 | 1 |
| 2.00 | 0.19 | 3.67 | 86 | 0.53 | 1 |
| 2.00 | 0.10 | 1.63 | 86 | 0.45 | 2 |
| 2.00 | 0.19 | 1.63 | 86 | 0.54 | 2 |
| 2.00 | 0.60 | 0.90 | 118 | 0.61 | 1 |
| 2.00 | 0.60 | 1.41 | 118 | 0.66 | 1 |
| 2.00 | 0.60 | 1.82 | 118 | 0.74 | 1 |

**Table A2.** *Cont.*

| Feature Values | | | | | True Label Values (1 = Flexure, 2 = Flexure–Shear, 3 = Shear) |
|---|---|---|---|---|---|
| *Aspect Ratio* | *Axial Load Ratio* | *$\rho_s$ (%)* | *$f_c$ (MPa)* | *$v_{max}/\sqrt{fc}$* | *Failure* |
| 2.00 | 0.35 | 1.41 | 118 | 0.67 | 1 |
| 2.00 | 0.35 | 1.82 | 118 | 0.67 | 1 |
| 7.64 | 0.34 | 0.12 | 40.6 | 0.13 | 1 |
| 6.04 | 0.50 | 3.15 | 72.1 | 0.19 | 1 |
| 6.04 | 0.36 | 2.84 | 71.7 | 0.19 | 1 |
| 6.04 | 0.50 | 2.84 | 71.8 | 0.19 | 1 |
| 6.04 | 0.50 | 5.12 | 71.9 | 0.19 | 1 |
| 6.04 | 0.45 | 4.02 | 101.8 | 0.21 | 1 |
| 6.04 | 0.46 | 6.74 | 101.9 | 0.21 | 1 |
| 6.04 | 0.45 | 2.72 | 102 | 0.18 | 1 |
| 6.04 | 0.47 | 4.29 | 102.2 | 0.19 | 1 |
| 4.70 | 0.43 | 1.00 | 34 | 0.27 | 1 |
| 4.70 | 0.43 | 2.00 | 34 | 0.27 | 1 |
| 4.70 | 0.20 | 2.00 | 34 | 0.23 | 1 |
| 4.70 | 0.46 | 1.33 | 34 | 0.29 | 1 |
| 4.70 | 0.46 | 2.66 | 34 | 0.33 | 1 |
| 4.70 | 0.46 | 2.66 | 34 | 0.31 | 1 |
| 4.70 | 0.46 | 1.26 | 34 | 0.30 | 1 |
| 4.70 | 0.23 | 1.26 | 34 | 0.28 | 1 |
| 4.70 | 0.46 | 1.26 | 34 | 0.31 | 1 |
| 4.70 | 0.46 | 2.66 | 34 | 0.33 | 1 |
| 3.00 | 0.05 | 1.00 | 69.6 | 0.20 | 1 |
| 3.00 | 0.05 | 1.00 | 69.6 | 0.20 | 1 |
| 3.00 | 0.10 | 1.00 | 67.8 | 0.28 | 1 |
| 3.00 | 0.10 | 1.00 | 67.8 | 0.28 | 1 |
| 3.00 | 0.21 | 1.00 | 65.5 | 0.32 | 1 |
| 3.00 | 0.21 | 1.00 | 65.5 | 0.31 | 1 |
| 3.00 | 0.00 | 1.00 | 37.9 | 0.23 | 1 |
| 3.00 | 0.00 | 1.00 | 37.9 | 0.23 | 1 |
| 3.00 | 0.14 | 1.00 | 48.3 | 0.25 | 1 |
| 3.00 | 0.14 | 1.00 | 48.3 | 0.25 | 1 |
| 3.00 | 0.36 | 1.00 | 38.1 | 0.33 | 1 |
| 3.00 | 0.36 | 1.00 | 38.1 | 0.33 | 1 |
| 3.50 | 0.11 | 0.76 | 24.9 | 0.31 | 1 |
| 3.50 | 0.16 | 0.76 | 26.7 | 0.32 | 1 |
| 3.50 | 0.22 | 0.76 | 26.1 | 0.37 | 1 |
| 3.50 | 0.11 | 0.73 | 25.3 | 0.31 | 1 |
| 3.50 | 0.16 | 0.73 | 27.1 | 0.34 | 1 |
| 3.50 | 0.21 | 0.73 | 26.8 | 0.37 | 1 |
| 3.50 | 0.11 | 0.71 | 26.38 | 0.31 | 1 |
| 3.50 | 0.15 | 0.71 | 27.48 | 0.34 | 1 |
| 3.50 | 0.21 | 0.71 | 26.9 | 0.36 | 1 |
| 2.67 | 0.00 | 0.04 | 21.9 | 0.23 | 3 |
| 1.33 | 0.00 | 0.09 | 16 | 0.38 | 3 |
| 3.92 | 0.00 | 0.96 | 102.7 | 0.20 | 1 |
| 3.92 | 0.20 | 0.96 | 86.3 | 0.34 | 1 |
| 3.92 | 0.00 | 0.96 | 87.5 | 0.19 | 1 |
| 3.92 | 0.10 | 0.96 | 83.4 | 0.26 | 1 |
| 3.92 | 0.20 | 0.96 | 90 | 0.30 | 1 |

**Table A2.** *Cont.*

| | Feature Values | | | | True Label Values (1 = Flexure, 2 = Flexure–Shear, 3 = Shear) |
|---|---|---|---|---|---|
| *Aspect Ratio* | *Axial Load Ratio* | *$\rho_s$ (%)* | *$f_c$ (MPa)* | *$v_{max}/\sqrt{fc}$* | *Failure* |
| 3.92 | 0.00 | 0.96 | 67.5 | 0.21 | 1 |
| 3.92 | 0.10 | 0.96 | 74.6 | 0.26 | 1 |
| 3.92 | 0.20 | 0.96 | 81.8 | 0.27 | 1 |
| 3.92 | 0.20 | 0.77 | 75.8 | 0.28 | 1 |
| 3.92 | 0.20 | 0.64 | 87 | 0.29 | 1 |
| 3.92 | 0.20 | 0.54 | 71.2 | 0.27 | 1 |
| 3.22 | 0.15 | 0.25 | 21.1 | 0.33 | 2 |
| 3.22 | 0.61 | 0.25 | 21.1 | 0.37 | 2 |
| 3.22 | 0.15 | 0.25 | 21.8 | 0.30 | 2 |
| 6.56 | 0.14 | 2.50 | 92.4 | 0.14 | 1 |
| 6.56 | 0.28 | 2.50 | 93.3 | 0.18 | 1 |
| 6.56 | 0.39 | 2.50 | 98.2 | 0.21 | 1 |
| 6.56 | 0.14 | 1.16 | 94.8 | 0.12 | 1 |
| 6.56 | 0.26 | 1.16 | 97.7 | 0.18 | 1 |
| 6.56 | 0.37 | 1.16 | 104.3 | 0.19 | 1 |
| 6.56 | 0.40 | 2.50 | 78.7 | 0.21 | 1 |
| 6.56 | 0.41 | 2.50 | 109.2 | 0.22 | 1 |
| 6.56 | 0.35 | 1.93 | 109.5 | 0.20 | 1 |
| 6.56 | 0.37 | 1.33 | 104.2 | 0.21 | 1 |
| 6.56 | 0.53 | 1.93 | 104.5 | 0.21 | 1 |
| 6.56 | 0.51 | 2.50 | 109.4 | 0.22 | 1 |
| 2.25 | 0.08 | 0.57 | 33.7 | 0.42 | 1 |
| 2.25 | 0.08 | 0.57 | 33.7 | 0.42 | 1 |
| 2.25 | 0.09 | 1.64 | 32.1 | 0.44 | 1 |
| 2.25 | 0.09 | 1.64 | 32.1 | 0.44 | 1 |
| 2.25 | 0.10 | 0.82 | 29.9 | 0.45 | 1 |
| 2.25 | 0.10 | 0.82 | 29.9 | 0.45 | 1 |
| 2.25 | 0.10 | 1.09 | 27.4 | 0.47 | 1 |
| 2.25 | 0.10 | 1.09 | 27.4 | 0.47 | 1 |
| 2.25 | 0.16 | 0.82 | 36.4 | 0.47 | 1 |
| 2.25 | 0.16 | 0.82 | 36.4 | 0.47 | 1 |
| 2.25 | 0.08 | 1.09 | 34.9 | 0.42 | 1 |
| 2.25 | 0.08 | 1.09 | 34.9 | 0.42 | 1 |
| 2.25 | 0.08 | 1.09 | 36.5 | 0.42 | 1 |
| 2.25 | 0.08 | 1.09 | 36.5 | 0.42 | 1 |
| 2.50 | 0.30 | 0.59 | 37.6 | 0.52 | 1 |
| 2.50 | 0.60 | 0.59 | 37.6 | 0.49 | 1 |
| 2.00 | 0.57 | 0.99 | 39.2 | 0.55 | 1 |
| 2.00 | 0.57 | 0.99 | 39.2 | 0.59 | 1 |
| 2.14 | 0.59 | 0.99 | 32.2 | 0.67 | 1 |
| 3.11 | 0.03 | 0.23 | 35.9 | 0.16 | 1 |
| 3.11 | 0.03 | 0.23 | 35.7 | 0.15 | 1 |
| 3.11 | 0.03 | 0.23 | 34.3 | 0.16 | 1 |
| 3.11 | 0.03 | 0.23 | 33.2 | 0.17 | 1 |
| 3.11 | 0.03 | 0.23 | 36.8 | 0.16 | 1 |
| 3.11 | 0.03 | 0.23 | 35.9 | 0.18 | 1 |
| 3.49 | 0.20 | 1.85 | 64.1 | 0.35 | 1 |
| 3.49 | 0.33 | 1.85 | 62.1 | 0.40 | 1 |
| 3.49 | 0.22 | 1.48 | 62.1 | 0.36 | 1 |

**Table A2.** *Cont.*

| Feature Values | | | | | True Label Values (1 = Flexure, 2 = Flexure–Shear, 3 = Shear) |
|---|---|---|---|---|---|
| *Aspect Ratio* | *Axial Load Ratio* | $\rho_s$ (%) | $f_c$ (MPa) | $v_{max}/\sqrt{fc}$ | *Failure* |
| 3.49 | 0.32 | 1.48 | 62.1 | 0.40 | 1 |
| 3.49 | 0.20 | 1.23 | 64.1 | 0.34 | 1 |
| 3.49 | 0.20 | 1.23 | 64.1 | 0.34 | 1 |

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
