# Peer review of "Random Forests Machine Learning Applied to PEER Structural Performance Experimental Columns Database"

_applsci, doi:10.3390/app132312821_

Round 1

Reviewer 1 Report

Comments and Suggestions for Authors

(1) Increase the number of recent references, ensuring that a significant portion of the cited literature is within the past three years to enhance the relevance and currency of the paper.

(2) Revise the abstract for logical coherence, particularly addressing the discrepancy between mentioning "incorporating physics knowledge into machine learning" without details in the paper. Provide a clear evaluation of the performance results of machine learning methods on datasets, specifying the method used and its accuracy.

(3) Specify the machine learning algorithm used when referring to 'supervised machine learning' in the title, providing clarity on the specific approach adopted.

(4) Streamline the description of the database, minimizing redundant content, and presenting essential information at a higher level.

(5) Enhance the description of the random forest algorithm's application process. Provide more details, such as the rationale behind choosing 1000 as the number of hyperparameter trees and how values for other important hyperparameters are determined.

(6) Include a data summary description table in the article to facilitate quick understanding for readers, specifying feature values, predicted label values, and the classification nature of the algorithm.

(7) Address the low precision and recall scores for the prediction of 'Shear.' Propose optimizations and improvements to enhance these scores, providing a more robust analysis.

(8) The authors may add more state-of-art application articles for the integrity of the manuscript (Exploring temperature-resilient recycled aggregate concrete with waste rubber: An experimental and multi-objective optimization analysis; Reviews on Advanced Materials Science. An experimental investigation and machine learning-based prediction for seismic performance of steel tubular column filled with recycled aggregate concrete; Reviews on Advanced Materials Science).

Author Response

Answers to Reviewer’s 1 Comments can be found in the attached file.

Reviewer 2 Report

Comments and Suggestions for Authors

Comments

This paper aims to incorporate “physical knowledge” into machine learning methods. For that purpose, ML methods are utilized to determine the failure mode of reinforced concrete columns. The conclusion is that the random forests is very accurate on identifying the failure mode of the experimental data.

Considering this, the reviewer has following comments prior to possible publication.

1.      The abstract does not make sense at all. The authors highlighted that existing methodologies fail to incorporate a so called “physical meaning”. The reviewer is confused about what that means. And later the authors’ objective is completely different that is studying failure modes.

2.      Also, please highlight the novelty of your work as much of the research has been conducted in this area and the reviewer is confused about the contribution of this paper.

3.      PEER data is publically available and doing some machine learning is not a significant contribution. Also what is “physical meaning” in the failure mode estimation of RC columns?

4.      Till figure 5, its just representation of the data in terms of Box plot. Also the quality of figures should be improved significantly.

5.      How many models are utilized to arrive at the conclusion that random forest is best ?

6.      Also, the reviewer doubts that methodology presented here is rigorous enough and hence the authors claims that random forest is best is rather doubtful?

7.      The literature review is also incomplete.

Comments on the Quality of English Language

Minor editing of English language required

Author Response

Answers to Reviewer’s 2 Comments can be found in the attached file.

Round 2

Reviewer 2 Report

Comments and Suggestions for Authors

The authors have successfully answered the comments of the reviewer

Comments on the Quality of English Language

Minor editing of English language required